# Generation of endogenous pH-sensitive EGF receptor and its application in high-throughput screening for proteins involved in clathrin-mediated endocytosis

Mads Breum Larsen[1], Mireia Perez Verdaguer[1], Brigitte F Schmidt[2], Marcel P Bruchez[2,3,4,5], Simon C Watkins[1]*, Alexander Sorkin[1]*

[1]Department of Cell Biology, School of Medicine, University of Pittsburgh, Pittsburgh, United States; [2]Molecular Biosensor and Imaging Center, Carnegie Mellon University, Pittsburgh, United States; [3]Department of Biological Sciences, Carnegie Mellon University, Pittsburgh, United States; [4]Department of Chemistry, Carnegie Mellon University, Pittsburgh, United States; [5]Sharp Edge Laboratories, Pittsburgh, United States

**Abstract** Previously we used gene-editing to label endogenous EGF receptor (EGFR) with GFP and demonstrate that picomolar concentrations of EGFR ligand drive signaling and endocytosis of EGFR in tumors in vivo (Pinilla-Macua et al., 2017). We now use gene-editing to insert a fluorogen activating protein (FAP) in the EGFR extracellular domain. Binding of the tandem dye pair MG-Bis-SA to FAP-EGFR provides a ratiometric pH-sensitive model with dual fluorescence excitation and a single far-red emission. The excitation ratio of fluorescence intensities was demonstrated to faithfully report the fraction of FAP-EGFR located in acidic endosomal/lysosomal compartments. Coupling native FAP-EGFR expression with the high method sensitivity has allowed development of a high-throughput assay to measure the rates of clathrin-mediated FAP-EGFR endocytosis stimulated with physiological EGF concentrations. The assay was utilized to screen a phosphatase siRNA library. These studies highlight the utility of endogenous pH-sensitive FAP-receptor chimeras in high-throughput analysis of endocytosis.
DOI: https://doi.org/10.7554/eLife.46135.001

*For correspondence:
swatkins@pitt.edu (SCW);
sorkin@pitt.edu (AS)

## Introduction

Ligand binding to EGFR at the cell surface leads to activation of the receptor tyrosine kinase activity, initiation of multiple signaling cascades and ultimately cell proliferation, differentiation or oncogenic transformation (reviewed in *Lemmon and Schlessinger, 2010*). Ligand binding also results in rapid endocytosis of EGFR and subsequent degradation of the receptor in lysosomes (reviewed in *Sorkin and Goh, 2009a*). Endocytosis and lysosomal degradation are thought to be the major regulators of EGFR signaling (reviewed in *Sigismund et al., 2012*; *Sorkin and von Zastrow, 2009b*). However, the molecular mechanisms of EGFR endocytosis are not well understood. Multiple pathways of EGFR endocytosis have been documented including clathrin-mediated (CME) (*Gorden et al., 1978*; *Hanover et al., 1984*; *Motley et al., 2003*) and clathrin-independent endocytosis (for example, *Caldieri et al., 2017*; *Orth et al., 2006*). While receptor ubiquitination by the Cbl E3 ubiquitin ligase has been proposed to be one of the redundant mechanisms of the EGFR CME (*Fortian et al., 2015*), EGFR internalization was shown to be normal in mouse embryonic fibroblasts derived from a double-knockout of c-Cbl and Cbl-b (*Mohapatra et al., 2013*). Further, despite the well-established importance of the receptor kinase activity and phosphorylation in EGFR CME

(reviewed in *Sorkin and von Zastrow, 2009b*), phosphatases involved in the endocytic process have not been identified. These deficiencies and inconsistencies in our understanding of EGFR endocytosis can be, in large part, explained by the unusual dose-dependence of the various EGFR endocytic mechanisms, their apparent redundancies and the experimental conditions used. Specifically, the use of overexpressed recombinant EGFR and high ligand concentrations favor EGFR internalization through clathrin-independent rather than CME pathways. To resolve these experimental inconsistences, we were prompted to renew our efforts to systematically analyze proteins and processes involved in fundamental mechanisms underlying EGFR trafficking.

Our recent study using endogenously expressed GFP-tagged EGFR allowed the demonstration that low concentrations of EGFR ligands are sufficient to drive EGFR-dependent growth of mouse tumor xenografts and that EGFR endocytosis in tumors in vivo is clathrin-mediated (*Pinilla-Macua et al., 2017*). These results inspired us to develop a more sophisticated endocytosis assay that would allow a quantitative analysis of the receptor traffic under physiological conditions. To implement this, we used CRISPR/Cas9 gene-editing to generate a new functional fusion protein of endogenous EGFR that can be used in an endocytosis assay that reports the pH difference between extracellular and endosomal pH. We took advantage of a pH-sensitive, membrane-impermeant tandem dye MG-Bis-SA that becomes fluorescent only when bound to the fluorogen activating protein (FAP) and whose dual fluorescence excitation results in pH-dependent far-red ratiometric emission (*Perkins et al., 2018b*). In the latter study, the effectiveness of MG-Bis-SA for studying endocytosis and recycling of FAP-tagged β2-adrenergic receptor (B2AR) overexpressed in HEK293 cells at single-endosome level was demonstrated. We have previously developed FAP-tagged antibodies and nanobodies to EGFR to study its expression and endocytosis in cells with high receptor levels (*Ackerman et al., 2018*; *Tan et al., 2017*; *Wang et al., 2017*; *Wang et al., 2015*). However, combining the MG-Bis-SA/FAP-based methodology and cells expressing endogenous EGFR with FAP at the amino-terminus allowed the development of an experimental model system that is amenable to a simple and highly sensitive single-cell readout, although in a high-throughput assay format, such that we can monitor and measure constitutive or ligand-induced EGFR endocytosis under physiological conditions, for example using low EGFR ligand concentrations.

## Results and discussion

To generate the pH-sensitive EGFR, FAP was inserted at the amino-terminus of the EGFR molecule in the *EGFR* gene locus downstream of the sequence encoding the signal peptide and upstream of the sequence of the mature EGFR using CRISPR/Cas9 gene-editing method (*Figure 1A*). Genome editing was performed in HeLa cells because EGFR endocytosis and signaling have been extensively studied and well characterized in these cells by others and ourselves. Several single-cell clones expressing FAP-EGFR were selected, and the clone EE7 with the most homogenous expression of FAP-EGFR within the cell population was used for subsequent experiments. Western blotting analysis confirmed FAP-EGFR fusion and demonstrated that FAP was inserted in all three copies of the *EGFR* gene in the EE7 clone as no untagged EGFR was detected (*Figure 1B*). While this clone did express a higher level of EGFR compared to parental HeLa cells, the activity of FAP-EGFR in EE7 cells measured as receptor phosphorylation at Tyr1068 per ligand-occupied receptor was equivalent to parental HeLa cells (*Figure 1C and D*). The $^{125}$I-EGF internalization rates were high and also essentially similar in parental HeLa and EE7 cells (Ke = 0.49–0.51/min; *Figure 1E*). Such rates are within the range of typical internalization rates via clathrin-coated pits (reviewed in *Sorkin and Goh, 2009a*). The functionality of FAP-EGFR demonstrated in *Figure 1B–E* is consistent with the previous demonstration that the insertion of a large tag, such as YFP, in the amino-terminus of EGFR does not affect receptor activity and endocytosis (*Kozer et al., 2011*).

To examine ligand-induced endocytosis of FAP-EGFR by single-cell imaging, surface FAP-EGFR was labeled by a transient incubation of EE7 cells with membrane-impermeant fluorogen MG-B-Tau, a derivative of malachite green (*Yan et al., 2015*). The cells were further incubated with EGF-Rhodamine (EGF-Rh; 6 ng/ml) for 15 min at 37℃ to stimulate receptor endocytosis. *Figure 1F* shows strong accumulation of MG-B-Tau in endosomes where it is appropriately co-localized with EGF-Rh.

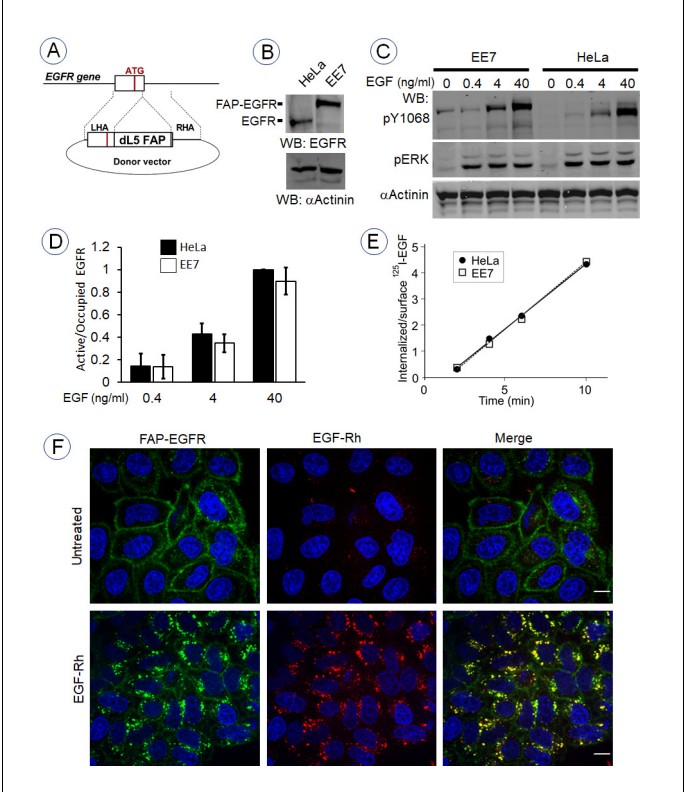

**Figure 1.** Generation and characterization of cells expressing endogenous FAP-tagged EGFR. (**A**) Schematics of the gene-editing of *EGFR* gene by inserting the FAP dL5 sequence between the sequence encoding the signal peptide and the mature EGFR. See 'Materials and methods' for description of sgRNAs and a donor vector. (**B**) EE7 clone of HeLa cells expressing FAP-EGFR and parental HeLa cells were lysed, lysates were electrophoresed in 6% acrylamide gel, and western blot analysis was performed with the EGFR and α-actinin (loading control) antibodies. (**C**) Parental HeLa and EE7 cells were serum-starved and stimulated with the range of EGF concentrations for 5 min at 37°C. Lysates were probed by western blotting using indicated antibodies. (**D**) Quantification of western blotting images exemplified in (**C**). The amount of EGFR or FAP-EGFR phosphorylated at Tyr1068 was determined by normalizing the pY1068 signal by the loading control and by the amount of ligand-occupied receptors in cells determined by incubating cells with $^{125}$I-EGF under conditions identical to those used in (**C**). Mean values from three independent experiments are presented (± SEM). (**E**) Internalization rates of 1 ng/ml $^{125}$I-EGF in parental HeLa and EE7 cells were measured as described in 'Materials and methods'. (**F**) Serum-starved EE7 cells were incubated for 1 min with MG-B-Tau (50 nM) and then the cells were further incubated in the absence or presence of 6 ng/ml EGF-Rh at 37°C for 15 min. Imaging was performed through the 561 nm (*red*, EGF-Rh) and 640 nm channels (*green*, MG-B-Tau). Nuclei were stained with Hoechst 33342 (*blue*). Scale bar, 10 μm.

DOI: https://doi.org/10.7554/eLife.46135.002

Together, the data in *Figure 1* validate EE7 cells expressing endogenous FAP-EGFR as an appropriate model to study EGFR endocytosis.

To develop and employ a quantitative internalization assay, EE7 cells were labeled with MG-Bis-SA, a tandem dye sensor consisting of a fluorogenic acceptor, MG, and a pH-dependent Cy3 FRET (fluorescence resonance energy transfer) donor (*Perkins et al., 2018b*). The properties of MG-Bis-SA and the principles of the ratiometric internalization assay using this dye are described in the latter study. In brief, the Cy3 excitation energy is transferred non-radiatively by FRET to MG; in solution MG acts as a quencher, whereas in FAP-bound state, the MG displays sensitized FRET emission (*Figure 2A*). At low pH, Cy3 (donor) absorption and energy transfer to MG are increased. Therefore, the excitation contribution of the pH-sensitive moiety (561 nm excitation) to far-red emission (680–700 nm) from the MG-FAP complex, increases at an acidic pH, whereas MG fluorescence (680–700 nm) by direct excitation at 640 nm is independent of pH. The ratio of the fluorescence emission intensities of MG-Bis-SA (680–700 nm) bound to FAP excited at 561 nm and 640 nm (termed as the

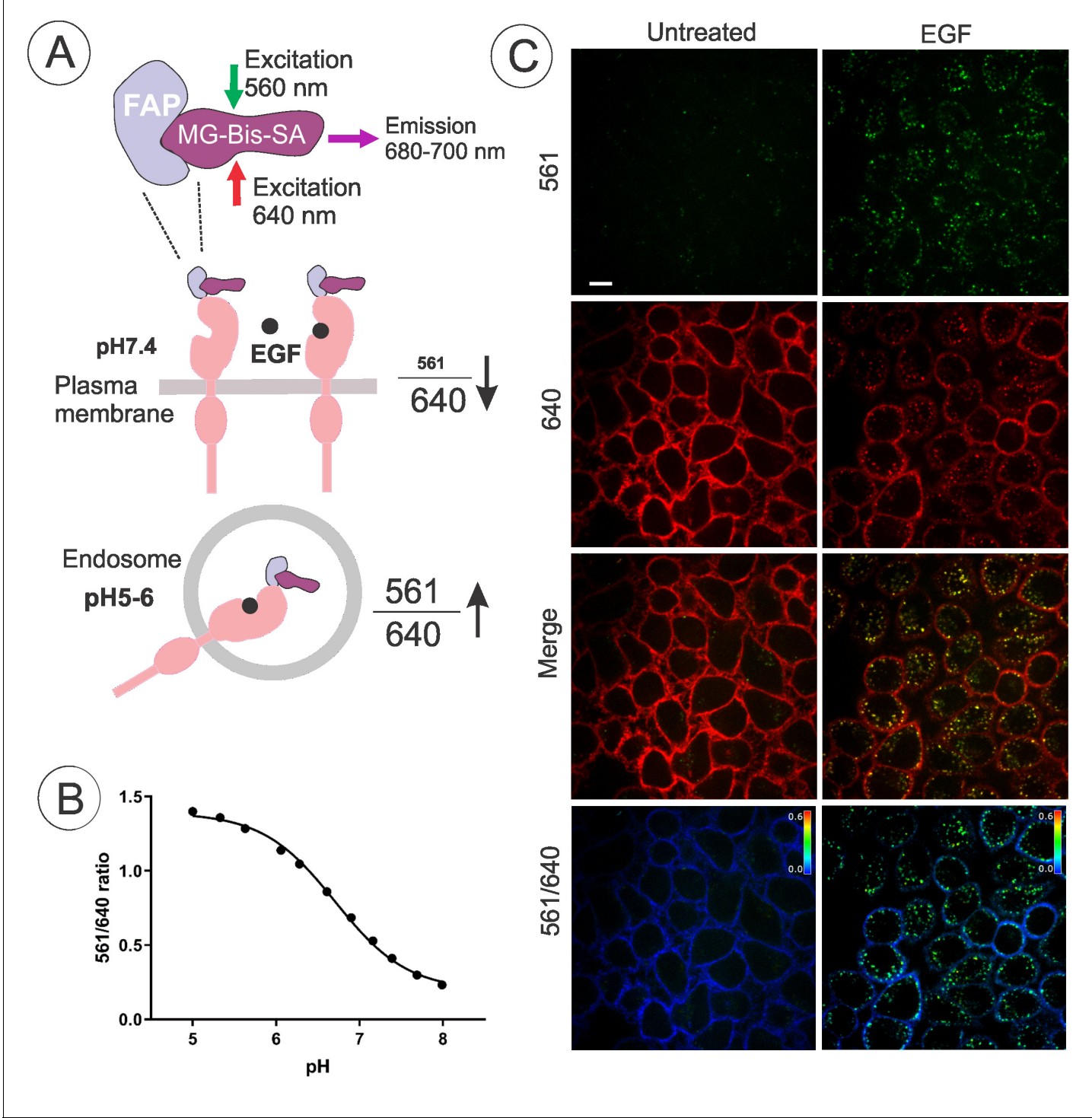

**Figure 2.** Quantitative analysis of FAP-EGFR endocytosis in EE7 cells. (**A**) The basis of the pH dependence of the 561/640 ratio of the fluorescence emission of MG-Bis-SA bound to FAP-EGFR and its use to measure the amount of FAP-EGFR in endosomes. Far-red emission of MG-Bis-SA results from the direct excitation at 640 nm (independent of pH) and the excitation of Cy3 at 561 nm and energy transfer to MG (dependent on pH and high at low pH). (**B**) pH dependence of the 561/640 ratio measured by labeling of surface-exposed FAP-EGFR in EE7 cells grown in 96-well plates with MG-Bis-SA and conducting fluorescence measurements in a series of buffers with different pH. Each data point is a mean of 7 wells (±S.D.) (**C**) An example of EGF-induced internalization of FAP-EGFR visualized using dual-excitation MG-Bis-SA imaging. EE7 cells were serum-starved and incubated with MG-Bis-SA (100 nM in DMEM) for 1 min, and then with 4 ng/ml EGF for 15 min at 37°C. 3-D live-cell imaging through the FRET channel (*green*, excitation 561 nm; emission 680 nm) and the 640 nm channel (*red*, excitation 640 nm; emission 680 nm) was performed. Individual confocal sections through the
*Figure 2 continued on next page*

*Figure 2 continued*

middle of the cells are presented. The 561/640 ratio image is presented as pseudocolored image modulated to the intensity of the 640 nm channel. All fluorescence intensities scales are identical between untreated and EGF-treated cells. Scale bar, 10 µm.

DOI: https://doi.org/10.7554/eLife.46135.003

'561/640' ratio) increases several fold at pH of endosomes (~pH 5.5) as compared to pH 7.4 of the medium (*Perkins et al., 2018b*). In our measurements in EE7 cells, the 561/640 ratio of MG-Bis-SA bound to FAP-EGFR was ~3 fold higher at pH 5.5 than at pH 7.4 (*Figure 2B*). Therefore, 561/640 ratio was used as a measure of the abundance of endosomal FAP-EGFR in subsequent experiments.

Various pH-sensitive probes have been previously developed for studying membrane trafficking. Tagging transmembrane proteins ('cargo') with pH-sensitive fluorescent proteins (FPs), such as pHluorin, has been widely used (reviewed in *Bizzarri et al., 2009*), but this approach has a major disadvantage due to its inability to distinguish biosynthetic and internalized pools of intracellular FP-

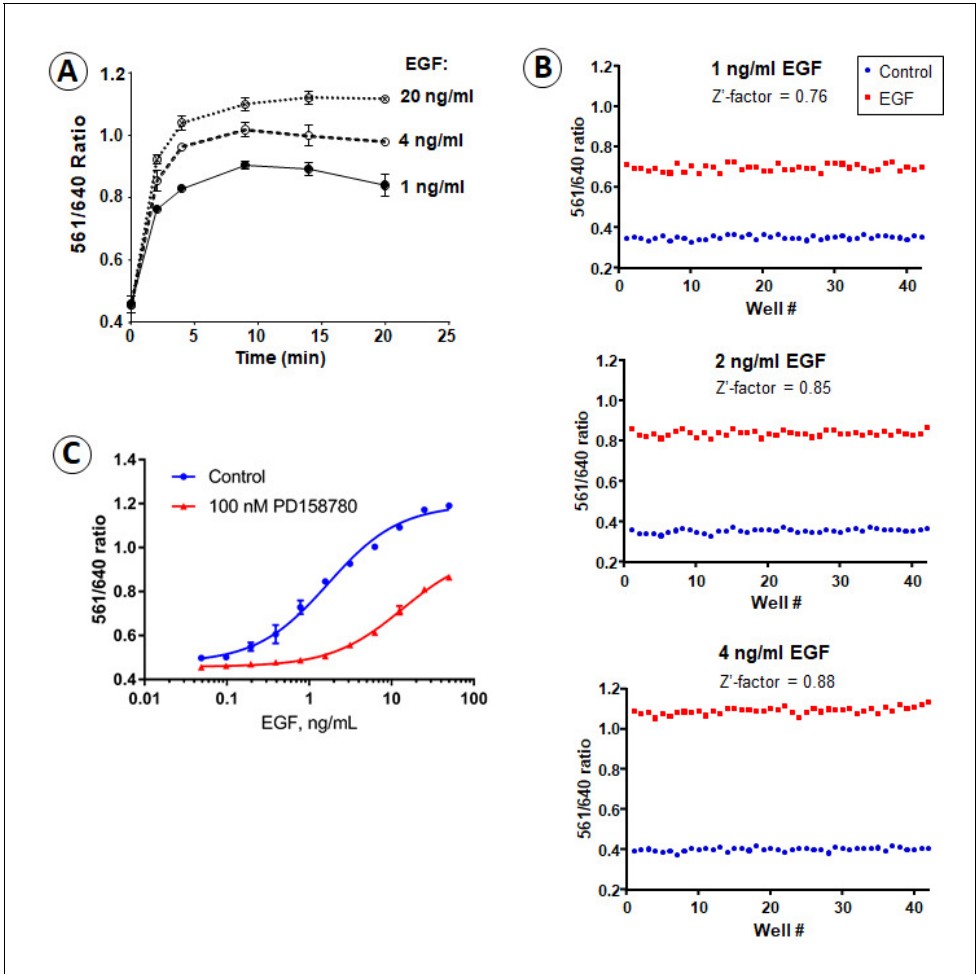

**Figure 3.** Time-course, dose-dependence and receptor-kinase-dependence of FAP-EGFR endocytosis. (A) Time-course of the 561/640 ratio measured in serum-starved EE7 cells grown in 96-well plates. The cells were labeled with MG-Bis-SA and incubated with 1, 4 or 20 ng/ml EGF at 37°C. The 561/640 ratio was measured as described in 'Material and methods'. (B) To test the robustness of the assay at physiologically relevant EGF concentrations, we calculated the Z'-factor from the assay run with half a plate run as control and the other half with the addition of 1, 2 or 4 ng/ml EGF. The Z'-factor was calculated to be 0.76–0.88, indicating the high robustness of the assay. (C) EE7 cells grown in 96-well plates were serum-starved, incubated with vehicle (DMSO) or PD158780 (50 nM) for 30 min, labeled with MG-Bis-SA and further incubated with the range of EGF concentrations for 15 min at 37°C in the same media. The 561/640 ratio was measured as described in 'Material and methods'.

DOI: https://doi.org/10.7554/eLife.46135.004

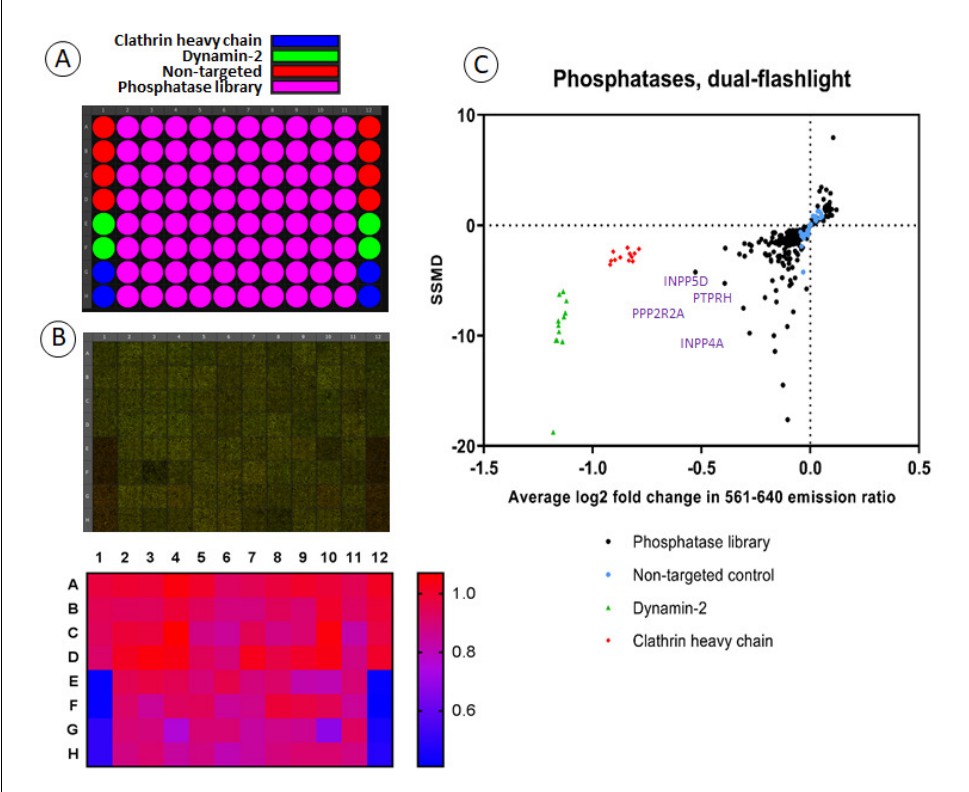

**Figure 4.** Screening a human phosphatase siRNA library for effects on FAP-EGFR internalization. (**A**) Each 96-well plate contained 80 siRNA pools targeting different phosphatases as well as eight negative control wells with non-targeted siRNA and four positive control wells each with siRNA targeting either clathrin heavy chain (CHC) or dynamin-2. Three plates covered the entire siRNA library of 240 phosphatases. (**B**) An example of the typical data resulting from this screen. EE7 cells were labeled with MG-Bis-SA and incubated with 1 ng/ml EGF for 15 min at 37˚C followed by ratiometric imaging as described in 'Materials and methods'. In the top panel, green is the signal from 561 nm excitation (FRET) and red is the signal from 640 nm excitation (MG) and as expected, a clear difference is seen between negative control wells and wells with knockdown of clathrin heavy chain or dynamin-2. The heatmap (bottom image) shows the calculated 561/640 ratio. (**C**) Summarized results from siRNA screening presented as a dual-flashlight plot. For each well, the average log2 fold change in the 561/640 emission ratio from the median of the negative controls of three independent screen repeats is plotted versus the strictly-standardized mean of the difference (SSMD). See *Figure 4—source data 1* for all values of the 561/640 ratio and SSMD obtained in three rounds of screening.

DOI: https://doi.org/10.7554/eLife.46135.005

The following source data is available for figure 4:

**Source data 1.** The data obtained in three independent screenings of the phosphatase siRNA library are presented.
DOI: https://doi.org/10.7554/eLife.46135.006

tagged cargo. A number of methods using pH-sensitive fluorophores that can be conjugated to genetically encoded tags, such as SNAP or Halo, were also applied to study membrane traffic (*Los and Wood, 2007*; *Martineau et al., 2017*; *Takeda et al., 2012*). However, such methods often require long incubations with the fluorophore dye, washing off unbound fluorophores and may result in substantial background fluorescence. Recently developed radiometric pH biosensors called SRpHi are excellent endosomal pH probes but they are not intended for labeling of endocytic cargo (*Richardson et al., 2017*). In the FAP-based approach, low backgrounds are achieved due to the 'dark' state of unbound MG. Moreover, labeling of surface-exposed FP-tagged cargo with MG-Bis-SA is virtually instantaneous, which allows monitoring the endocytic process from its initial steps at the plasma membrane and through the sorting in endosomes without an interference from constitutive endosomal and biosynthetic pools of the cargo.

An example of the ratiometric imaging of FAP-EGFR endocytosis in EE7 cells is presented in *Figure 2C*. The cells were labeled with MG-Bis-SA for 1 min and then incubated without or with EGF (4 ng/ml) for 15 min at 37˚C. Strong FRET signal (excitation at 561 nm, emission at 680 nm) was

observed in the vesicular compartments of cells stimulated with EGF but not in non-stimulated cells. By contrast, a strong far-red fluorescence excited at 640 nm with minimal FRET signal was detected at the plasma membrane of non-stimulated cells, primarily at the cell edges, ruffles and protrusions; this signal was clearly reduced in EGF-stimulated cells. This single-cell analysis shows an overall increase in the 561/640 ratio, clearly demonstrating that a quantitative analysis of FAP-EGFR endocytosis is feasible using this tool.

Subsequently, we used 96-well plates to measure the rates of FAP-EGFR endocytosis in a high-throughput format using the ratiometric assay. Cells were grown to confluence and serum-starved overnight, surface-exposed FAP-EGFR was labeled with impermeant MG-Bis-SA, and endocytosis was stimulated by EGF. The value of the 561/640 ratio increased during the first 10–15 min after EGF (1–20 ng/ml) stimulation at 37°C reaching a plateau at 15 min (*Figure 3A*). Therefore, the 15 min EGF incubation timepoint, the condition of a maximally high accumulation of FAP-EGFR in endosomal compartments, was used to measure the 561/640 ratio in subsequent high-throughput experiments.

Concentrations of EGFR ligands accessible to EGFR in mammalian tissues are typically below 1–2 ng/ml (*Connolly and Rose, 1988*; *Dvorak, 2010*; *Hirata and Orth, 1979*; *Ishikawa et al., 2005*; *Murdoch-Kinch et al., 2011*; *Oka and Orth, 1983*; *Rich et al., 2017*), and our recent studies suggested that concentration of EGFR ligands of 1 ng/ml or less are present in mouse tumor xenografts and are sufficient to support EGFR-dependent tumor growth (*Pinilla-Macua et al., 2017*). We confirmed using the ratiometric endocytosis assay that the extent of FAP-EGFR endocytosis can be measured with a high statistical significance over a range of EGF concentrations, including 1 ng/ml, in the 96-well plate format (*Figure 3B*). Importantly, endocytosis of FAP-EGFR stimulated with low EGF concentrations was blocked by the inhibitor of EGFR kinase activity (PD158780), whereas a significant component of kinase-insensitive FAP-EGFR endocytosis was observed when high, saturating concentrations of EGF were used (*Figure 3C*). These data are consistent with the concept that EGFR tyrosine kinase activity is important at low concentrations of EGF (conditions favoring internalization through the CME pathway) but not at high EGF concentrations (conditions favoring clathrin-independent endocytosis) (*Chen et al., 1989*; *Honegger et al., 1987*; *Lund et al., 1990*; *Sorkina et al., 2002*; *Wiley, 1988*). Altogether, the data presented in *Figure 3* demonstrate that the MG-Bis-SA/FAP-EGFR assay in 96-well plates is sufficiently sensitive and reliable to be used for the analysis of the mechanism of the EGFR CME.

To test the applicability of the experimental model of FAP-EGFR expressing EE7 cells and the ratiometric internalization assay to a high-throughput analysis, we performed screening of a siRNA library targeting 240 human phosphatases. Each 96-well plate included 80 siRNA pools targeted to phosphatases, negative controls (non-targeting siRNAs), and siRNAs to clathrin heavy chain (CHC) and dynamin-2, proteins essential for CME (positive controls) (*Figure 4A*). The same siRNAs targeting CHC and dynamin-2 were shown to block internalization of 1 ng/ml $^{125}$I-EGF in our early studies (*Huang et al., 2004*). To identify phosphatases involved specifically in the CME of EGFR, the cells labeled with MG-Bis-SA were stimulated with 1 ng/ml EGF for 15 min at 37°C, and the 561/640 ratio was measured in four microscopic fields for each well as described in 'Material and methods' (*Figure 4B*). Importantly, the use of a high-magnification objective lens allowed real-time visual inspection of potential aberrant cell morphologies in individual wells of interest. As shown in the dual-flashlight plot that represents a summary of three independent repeats of screening experiments (*Figure 4C*), FAP-EGFR endocytosis was strongly inhibited by CHC and dynamin-2 depletion, confirming that under conditions of this assay the endocytosis is clathrin-mediated. Several siRNA pools to phosphatases were found to significantly inhibit FAP-EGFR endocytosis (*Figure 4C* and *Figure 4—source data 1*). Interestingly, among phosphatases identified in the screen as potentially important for EGFR endocytosis, are several phosphotidyl-inositide phosphatases, such as IPDD5D (SHIP1) (*Figure 4C*), INDDL1 (SHIP2) (*Figure 4—source data 1*) and INDD4A (*Figure 4C*). All three phosphatases have been implicated in various endocytic processes (*Erneux et al., 2011*; *Kamen et al., 2007*; *Nigorikawa et al., 2015*), and SHIP2 has been previously shown to be important for CME (*Nakatsu et al., 2010*). Several other phosphatases identified in our screens as potential regulators of EGFR endocytosis, such as PTPRH (*Figure 4C*), have been implicated in dephosphorylation of EGFR (*Yao et al., 2017*). The precise function of all above-mentioned phosphatases in EGFR endocytosis is unknown and will be a subject of our continuing investigations.

The data presented in *Figure 4* highlight the effectiveness of the FAP-EGFR-based assay as a high-throughput screen for proteins important in the CME of EGFR. EGFR endocytosis has been previously analyzed using high-throughput screens of targeted and whole-genome siRNA libraries (*Collinet et al., 2010*; *Pelkmans et al., 2005*). These studies used labeled EGF to follow endocytosis of EGFR and a complexed multi-parameter single-cell image analysis to score for the effects of siRNAs. These screens were instrumental in identifying several new regulatory processes and proteins, although findings did not advance our understanding of the molecular mechanisms of EGFR endocytosis. Importantly, the downstream analysis in those screens required high concentrations of labeled EGF. By contrast, the screening approach using a combination of receptor tagging by gene-editing and ratiometric measurements described here is based on monitoring the localization of the endogenous receptor itself. Therefore, the FAP-based method allows exquisitely sensitive screening for regulators of the constitutive EGFR endocytosis, and EGFR endocytosis induced by other EGFR ligands, that are technically difficult to label. Also, as we are directly and selectively labeling surface EGFR, it can be used to study ligand-independent endocytosis, such as endocytosis induced by activated p38 MAP kinase (*Frey et al., 2006*; *Vergarajauregui et al., 2006*; *Zwang and Yarden, 2006*).

Constitutive endocytosis of EGFR has been previously studied by measuring the uptake rates of radiolabeled and biotinylated antibodies bound to the EGFR extracellular domain (*Burke et al., 2001*; *Burke and Wiley, 1999*). Such methods, especially using radiolabeled antibody, are highly sensitive, although their implementation in a high-throughput assay format is technically difficult. Further, antibody-based measurements can be affected by the inability of an antibody to recognize receptors irrespective of their conformation and ligand occupancy, and by the bi-valent IgG binding; and these assays involve a 'terminal' antibody-stripping step at 4°C at each point of a time-course. Imaging-based FAP-EGFR methodology, besides its demonstrated amenability to a high-throughput screening, adds extra dimensions to the receptor endocytosis analysis. First, it allows quantitative monitoring of the entire time-course of FAP-EGFR internalization in a single living cell or a group of cells because no washing and stripping are required. Second, FAP-EGFR live-cell imaging can be performed on a pixel-by-pixel basis at high resolution, and thus can report the subcellular localization of the receptor in living cells. Therefore, the FRET signal (561 nm excitation) can be measured specifically in the endosome/lysosome compartments containing FAP-EGFR rather than in an entire cell. The fold-increase of the endolysosomal 561-nm-excited fluorescence of MG-Bis-SA bound to FAP-EGFR during endocytosis of this complex is significantly higher that the fold-increase of the total cellular fluorescence at 561 nm excitation (*Figure 2C*). Thus, the endosomal 561 nm-excited fluorescence alone can be utilized as the measure of FAP-EGFR internalization if screening is performed in cells expressing high receptor levels. In such cells, the concentration of EGFR at the cell surface remains practically constant during endocytosis triggered by low ligand concentrations because only a very small fraction of surface-exposed receptors is internalized.

Development of the experimental system described here can be technically challenging and time consuming as it involves labeling of endogenous EGFR by gene-editing. Because various types of cells may display different kinetics and mechanisms of constitutive and stimuli-induced endocytosis (see for example *Burke and Wiley, 1999*; *Jiang et al., 2003*; *Mohapatra et al., 2013*), application of the MG-Bis-SA/FAP-EGFR approach would require a FAP knock-in in each cell line of interest. However, engineering the FAP-EGFR by CRISPR/Cas9 method in human cell lines will be dramatically facilitated by the availability of the donor construct, efficient gRNAs and other technical information obtained in our experiments with HeLa cells. We believe that using these tools it is feasible to generate a cell line expressing endogenous FAP-EGFR within 1 month.

In conclusion, as more examples of the application of FAP-based pH sensors to measure and analyze the endocytosis of various cargo in single cells are documented (e.g. *Emmerstorfer-Augustin et al., 2018*; *Holleran et al., 2013*; *Perkins et al., 2018a*; *Perkins et al., 2018b*; *Pratt et al., 2017*), the feasibility and utility of the high-throughput method described here for systems biology studies of the endocytic trafficking of different, endogenous transmembrane proteins, and especially, ligand-independent transporters, channels and other cargoes, will become increasingly important.

# Materials and methods

**Key resources table**

| Reagent type (species) or resource | Designation | Source or reference | Identifiers | Additional information |
|---|---|---|---|---|
| Gene (*Homo sapiens*) | EGFR | NA | Q504U8 | |
| Recombinant DNA reagent | PX459 V2.0 | Addgene | #62988 | |
| Recombinant DNA reagent | pUC18 | Thermo Fisher Scientific | SD0051 | |
| Genetic reagent (human) | Dharmacon siGENOME SMARTpool siRNA Library – Human Phosphatase | Thermo Fisher Scientific | G-003705, Lot 10161 | |
| Antibody | anti-EGFR (mouse monoclonal) | Millipore (Transduction Laboratories) | 05–104 | (1:1000) |
| Antibody | anti- pTyr1068 EGFR (mouse monoclonal) | Cell Signaling Technology | 2236 | (1:1000) |
| Antibody | anti-α-actinin (rabbit polyclonal) | Cell Signaling Technology | 3134 | (1:1000) |
| Antibody | anti-pERK1/2 (rabbit polyclonal) | Cell Signaling Technology | 9101 | (1:1000) |
| Antibody | IRDye-800 (goat anti mouse) | LI-COR | 926–32210 | (1:20000) |
| Antibody | IRDye-680 (goat anti mouse) | LI-COR | 926–32220 | (1:20000) |
| Chemical compound, drug | MG-B-Tau | Bruchez laboratory | *Yan et al., 2015* | |
| Chemical compound, drug | MG-Bis-SA | Bruchez laboratory | *Perkins et al., 2018b* | |
| Chemical compound, drug | PD158780 | Millipore (Calbiochem) | 513035 | |
| Chemical compound, drug | human recombinant EGF | BD Bioscience | 354052 | |
| Chemical compound, drug | mouse receptor-grade EGF | Corning | 354010 | |
| Chemical compound, drug | 125-iodine, carrier free | PerkinElmer | NEZ033A | |
| Chemical compound, drug | EGF-Rhodamine | Thermo Fisher Scientific | E3481 | |
| transfected construct (human) | siRNA to Dynamin 2 (SMARTpool) | Dharmacon/Thermo Fisher Scientific | *Huang et al., 2004* | |
| Transfected construct (human) | siRNA to Clathrin heavy chain (SMARTpool) | Dharmacon/Thermo Fisher Scientific | *Huang et al., 2004* | |
| Cell line (human) | HeLa | ATCC | CCL-2 | |
| Sofware | SlideBook6 | Intelligent-imaging Innovations, Inc | NA | |
| Sofware | Image Studio Lite | Li-COR, Inc | NA | |
| Software | NIS Elements | Nikon | NA | |

## Reagents and antibodies

Human recombinant EGF was from BD Bioscience. EGF-Rh was from Invitrogen. Monoclonal antibody to EGFR phosphotyrosine 1068 (pY1068), polyclonal antibody to α-actinin, polyclonal antibody to ERK1/2 and monoclonal antibody to phosphorylated ERK1/2 were from Cell Signaling Technology (Beverly, MA). EGFR monoclonal antibody 04–105 were from Transduction Laboratories. PD158780 was from Calbiochem. MG-B-Tau and MG-Bis-SA were synthesized by the Bruchez lab (*Perkins et al., 2018b*; *Yan et al., 2015*). All chemicals were from ThermoFisher unless stated otherwise.

## Cell culture

HeLa cells were grown in Dulbecco's Modified Eagle Medium (DMEM) containing 10% fetal bovine serum (Gibco, Life Bioscience), 10,000 U/ml Penicillin G and 10 mg/ml streptomycin. The identity of HeLa cells was confirmed by CTR genotyping.

## CRISPR/Cas9-mediated gene editing

To generate EGFR tagged with the fluorogen-activating protein dL5 (*Szent-Gyorgyi et al., 2013*) (FAP), FAP dL5 sequence was inserted in the coding region of the *EGFR* gene after the signal peptide by CRISPR/Cas9-mediated gene-editing. A gRNA target site was identified by using online software from ATUM bio, CHOPCHOP, the Broad Institute sgRNA design tool, and the MIT CRISPR Design tool. The gRNA sequence AAGGTAAGGGCGTGTCTCGC[CGG] (PAM site in brackets) was identified by all above-mentioned software and consequently selected. The gRNA sequence was inserted into the PX459 V2.0 plasmid digested with BbsI using annealed oligos gRNA-1 plus and gRNA-1 minus. The donor template was constructed in a pUC18 vector background. We used 300 bp homology arms with flanking gRNA target sites, which has been shown to facilitate a high HDR efficiency (*Zhang et al., 2017*). Small flexible linkers were inserted between dL5 and the EGFR on both the 5'- and 3'-terminal ends to minimize any potential interference of dL5 on EGFR function. The 5' and 3' homology arms were amplified from genomic DNA using primer pairs gRNA-1 5'_HA_fwd/5'-HA-rev and 3'_HA_fwd/gRNA-1 3'_HA_rev, respectively. dL5 was amplified from a plasmid template using the primer pair FAP_fwd/FAP_rev. The resulting PCR products were gel purified and assembled in combination with SphI/SacI digested pUC18 vector by a Gibson assembly reaction. The central PAM sequence in the assembled donor template was changed from CGG to CGT by site-directed mutagenesis using primer pair gRNA-1 mut F/gRNA-1 mut R to prevent Cas9 cleavage of gene-edited alleles. *Oligo sequences:*

gRNA-1 plus 5'-caccgAAGGTAAGGGCGTGTCTCGC-3'
gRNA-1 minus 5'-aaacGCGAGACACGCCCTTACCTTc-3'
gRNA-1 5'_HA_fwd
5'-cagtgccaagcttgcatgaaggtaagggcgtgtctcgccggGCCTCCGCCCCCCGCACG-3'
5'-HA-rev 5'-tgaaccgccaccaccCTCCAGAGCCCGACTCGCCGG-3'
FAP_fwd 5'-gctctggagggtggtggcggttcaCAGGCCGTCGTTACCC-3'
FAP_rev 5'-ccccgcctcccccGGAGAGGACGGTCAGCTG-3'
3'_HA_fwd
5'-gtcctctccgggggaggcgggggtggaCTGGAGGAAAAGAAAGGTAAGGGCGTGTCTCGC-3'
gRNA-1 3'_HA_rev
5'-tgattacgaattcgagctccggcgagacacgcccttaccttGCGGTTCGGGGCGCCGGA-3'
gRNA-1 mut F 5'-TGTCTCGCCGtCTCCCGCGCC-3'
gRNA-1 mut R 5'-CGCCCTTACCTTTCTTTTCCTCCAGTCC-3'

HeLa cells were plated in a six-well plate and transfected the next day with equal amounts of PX459 with gRNA and donor template using Transit-2020 transfection reagent, following the manufacturer's recommendations. The transfected cells were maintained, detached using a citric saline buffer and subjected to fluorescence-activated cell sorting (FACS) for dL5/MG fluorescence (640 nM excitation, 680 nM emission) after the addition of MG-B-Tau. The sorted cells were propagated, and FACS was used as above to transfer single cells with high dL5/MG fluorescence to 96-well plates to create clonal lines expressing FAP-EGFR.

## EGF receptor internalization 96-well assay

Gene-edited HeLa cells (EE7 clone) were plated into optical 96-well plates (Greiner bio-one µCLEAR) at 30% confluency. The bottom of the 96-well plates was pretreated with rainx to facilitate the use of a high NA water immersion objective. The next day, cells were washed twice with PBS and incubated in serum-free DMEM containing 0.1% BSA for 16–24 hr to diminish baseline EGFR activation. The following day, the media was aspirated, and the cells incubated for 1 min in DMEM containing 100 nM MG-Bis-SA pH to label surface-resident FAP-EGFR. Next, the cells were washed with PBS and incubated in prewarmed DMEM/0.1% BSA containing the indicated concentration of EGF in the tissue culture incubator for 15 min to allow for FAP-EGFR internalization. To stop the internalization, the media was aspirated and ice-cold HEPES-buffered solution (in mM, 140 NaCl, 5 KCl, 1 $MgCl_2$, 0.1 $CaCl_2$, 25 HEPES, pH 8) added to the cells to stop trafficking and magnify the difference between endosomal and surface-exposed receptors. The cells were kept on ice and microscopy commenced shortly thereafter. We used a Nikon ECLIPSE T$i$-E microscopy system with an A1 resonant scanner and a 1.15 NA 40x water immersion objective to capture and stitch four image fields of every well using dual 561 nm/640 nm laser excitation and single emission (700/75 nm filter). The Nikon NIS Elements HCA JOBS module automatically calculated the average intensities of the emission from 561 nm/640 nm excitation for each stitched image. Parental HeLa cells were used for background signal correction.

In experiments with PD158780, the cells were pretreated with the compound for 30 min before starting the assay, and PD158780 was present in all solutions during EGF treatment. For the EGF time course experiment, all cells were labeled with MG-Bis-SA at the same time and kept in DMEM/ 0.1% BSA until the addition of EGF at different time points. At the end of the experiment, all cells had their media aspirated and were incubated with ice-cold HEPES-buffered solution at the same time, followed by microscopy. For the MG-Bis-SA calibration curve, cells were plated, serum-starved and labeled with MG-Bis-SA as described above. Following the wash with PBS, buffer with incremental increase in pH value from pH 5 to pH 8 was added to each row of a 96-well plate and microscopy commenced as above (ion concentrations in buffer in mM, 140 NaCl, 5 KCl, 1 $MgCl_2$, 1 $CaCl_2$, 50 mM MES for pH <6.8 or 50 mM HEPES for pH >6.8).

## siRNA-mediated knockdown

We used plates 1–3 of a Dharmacon siGENOME SMARTpool siRNA Library – Human Phosphatase (G-003705, Lot 10161) to test the effect of the knockdown of 240 different phosphatases with the assay. Cells were reverse transfected in 96-well plates, using 20 nM final siRNA concentration and 0.2 µl Dharmafect one per well, following the manufacturers recommendations. 48 hr later, media was aspirated, the cells washed with PBS and incubated in serum-free DMEM containing 0.1% BSA for 16–24 hr to diminish baseline EGFR activation. 1 ng/mL EGF was used to stimulate and measure EGFR internalization as described above. Duplex pools for CHC and dynamin-2 were from Dharmacon and characterized previously (*Huang et al., 2004*). Non-targeting siRNA was from Dharmacon.

## Single-cell fluorescence microscopy

For live-cell imaging, glass coverslips with EE7 cells were mounted into the microscope chamber in 1 ml DMEM/0.1% BSA medium, placed onto the microscope stage adaptor, and z-stacks of confocal images were acquired using a spinning disk confocal imaging system based on a Zeiss Axio Observer Z1 inverted fluorescence microscope (with 63x Plan Apo PH NA 1.4), equipped with a computer-controlled Spherical Aberration Correction unit, Yokogawa CSU-X1, Photometrics Evolve 16-bit EMCCD camera, environmental chamber and piezo stage controller and lasers (405, 445, 488, 515, 561, and 640 nm), all controlled by SlideBook6 software (Intelligent Imaging Innovation, Denver, CO). 15 (unless indicated otherwise) serial two-dimensional confocal images at 400 nm intervals were acquired by exciting at 561 nm and 640 nm with the emission at 680 (70) nm for MG-Bis-SA and using a 561 nm channel for Rhodamine in the environmental chamber ensuring a constant temperature (37˚C), humidity and 5% $CO_2$ atmosphere throughout the duration of imaging. EGF-Rh or unlabeled EGF was injected into the stage chamber in large volume (0.2 ml), thus ensuring rapid distribution of the ligand. Z-plane change time was ~3 ms. All image acquisition settings were identical for experimental variants in each experiment.

The 561/640 ratio images are presented in a quantitative pseudocolor mode. The value of the ratio is displayed stretched between the low and high values, according to a temperature-based lookup table with blue (cold) indicating low values and red (hot) indicating high values. To eliminate the distracting data from regions outside of cells, the 640 nm channel was used as a saturation channel, and the 561/640 ratio images are displayed as 640 nm-channel intensity modulated images. In these images, ratio data with 640 nm values greater than the high threshold of the fluorescence intensity are displayed at full saturation, whereas data values below the low threshold are displayed with no saturation (i.e. black).

## Western blotting

To probe for EGFR, and phosphorylated EGFR and ERK1/2, cells in 12-well plates were serum-starved overnight, treated with EGF at 37°C for 5 min, lysed in Triton X-100/Glycerol/Hepes buffer in the presence of orthovanadate and sodium fluoride as described (*Pinilla-Macua et al., 2017*). The lysates were resolved by 6% or 10% SDS-PAGE followed by transfer to the nitrocellulose membrane. Western blotting was performed with appropriate primary and secondary antibodies conjugated to far-red fluorescent dyes (IRDye-680 and −800) followed by detection using an Odyssey LI-COR system. Quantifications were performed using LI-COR software.

## $^{125}$I-EGF binding and internalization

Mouse receptor-grade EGF (Collaborative Research Inc, Bedford, MA) was iodinated using a modified Chloramine T method as described previously (*Sorkin and Duex, 2010*). Binding and internalization rates of $^{125}$I-EGF at 37°C were measured and analyzed as described (*Sorkin and Duex, 2010*). Quantifications of the ligand-bound EGFRs per cell were performed as described (*Sorkin and Duex, 2010*).

## Statistical analysis

Statistical significance (p value) was calculated using unpaired two-tailed Student's t tests (Graph-Pad). The Z'-factor (*Zhang et al., 1999*) is a commonly accepted measure of the suitability and quality of an assay for HTS. It is defined by:

$$1 - ((3\sigma_{c+} + 3\sigma_{c-})/(|\mu_{c+} - \mu_{c-}|))$$

where $\sigma_{c+}$ and $\sigma_{c-}$ denote the standard deviation of the positive and negative controls, respectively, and $\mu_{c+}$ and $\mu_{c-}$ denote the averages of the positive and negative controls, respectively. An assay with a Z'-factor between 0.5 and 1 is considered excellent for HTS (*Zhang et al., 1999*). The data for the dual-flashlight plot was calculated as described (*Zhang, 2011*).

# Acknowledgements

A Sorkin and SC Watkins were supported by the NIH grant GM124186. Marcel P Bruchez was supported by NIH grants R01EB017268 and R01GM114075.

# Additional information

### Competing interests

Marcel P Bruchez: founder of Sharp Edge Laboratories, a company that licensed the FAP technology from CMU. The author has no other competing interests to declare. The other authors declare that no competing interests exist.

### Funding

| Funder | Grant reference number | Author |
|---|---|---|
| National Institutes of Health | GM124186 | Simon C Watkins Alexander Sorkin |
| National Institutes of Health | R01EB017268 | Marcel P Bruchez Simon C Watkins |

| National Institutes of Health | R01GM114075 | Marcel P Bruchez |
| | | Simon C Watkins |

The funders had no role in study design, data collection and interpretation, or the decision to submit the work for publication.

## Author contributions
Mads Breum Larsen, Conceptualization, Data curation, Software, Formal analysis, Validation, Investigation, Visualization, Methodology, Writing—review and editing; Mireia Perez Verdaguer, Software, Formal analysis, Validation, Investigation, Visualization, Methodology, Writing—review and editing; Brigitte F Schmidt, Resources; Marcel P Bruchez, Resources, Writing—review and editing; Simon C Watkins, Conceptualization, Resources, Supervision, Writing—review and editing; Alexander Sorkin, Conceptualization, Software, Formal analysis, Supervision, Funding acquisition, Visualization, Writing—original draft, Project administration, Writing—review and editing

## Author ORCIDs
Marcel P Bruchez (iD) http://orcid.org/0000-0002-7370-4848
Simon C Watkins (iD) https://orcid.org/0000-0003-4092-1552
Alexander Sorkin (iD) https://orcid.org/0000-0002-4446-1920

## Decision letter and Author response
Decision letter https://doi.org/10.7554/eLife.46135.010
Author response https://doi.org/10.7554/eLife.46135.011

# Additional files

## Supplementary files
• Transparent reporting form
DOI: https://doi.org/10.7554/eLife.46135.007

## Data availability
All source data from library screening are provided in Figure 4—source data 1.

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
