## [Decision Letter]

Thank you for submitting your article "Endogenous pH-sensitive EGF receptor and its application in high-throughput analysis of clathrin-mediated endocytosis" for consideration by *eLife*. Your article has been favorably reviewed by two peer reviewers, and the evaluation has been overseen by John Kuriyan as the Reviewing Editor and Senior Editor. The following individual involved in review of your submission has agreed to reveal his identity: H Steven Wiley (Reviewer #1).

The reviewers have discussed the reviews with one another and the Reviewing Editor has drafted this decision to help you prepare a revised submission. You will see that no new experiments are asked for, and so we do not anticipate any difficulties in revising the paper.

Summary:

This is a nice technical advance that describes a novel approach for high-throughput screening of EGFR endocytosis. Larsen et al. report development and characterization of an assay to measure the rate of clathrin-mediated endocytosis (CME) of EGFR using HeLa cells in which CRISPR has been used to replace endogenous EGFR with a form of EGFR that can be labeled with a pH-sensitive fluorophore. The controls show that this tagging (FAP tagging) does not impact the internalization or physiological activity of the EGFR. In addition, they show that the sensitivity of the ratiometric technique is sufficient to allow assessment of physiological levels of ligand, and thus can be used to follow relevant mechanisms that regulate trafficking under physiological conditions.

This assay closely resembles a similar assay developed using transfection of a labeled β2-adrenergic receptor in HEK293, which was published by the Bruchez group in Biochemistry in 2018. The authors employ this assay in an siRNA screen of 240 phosphatases to identify phosphatases involved in CME of EGFR. At least three phosphotidyl-inositide phosphatases along with several phosphatases previously implicated in dephosphorylation of EGFR were identified in this screen. The precise roles of these phosphatases in CME of EGFR will be the subject of future studies.

On balance, the FAP-tagging approach seems ideally suited for high-throughput screening experiments because of its ability to use a single emission wavelength for the readout, the uniformity of the labeling, the relatively high S/N ratio that permits the use of low ligand concentrations and the ability to follow both empty and occupied receptors. The clean and consistent results obtained with both clathrin and dynamic siRNA pools is impressive (Figure 4). For screens of molecules that perturb the endocytosis of the EGFR (or other surface-associated proteins that can be tagged), this is certainly a powerful approach. It also shows the power of CRISPR-mediated tagging approaches and is likely to inspire the application of this technology to similar problems in biology.

Essential revisions:

Please discuss the following issues, to place the work in better context, noting the comments made by the reviewers below.

Please discuss the earlier work by others on the β2-adrenergic receptor more clearly, showing that the present work is an extension of work that had already developed the basic technology. We recognize that the advance here is to tag the endogenous receptor.

Although this is clearly a very powerful technological advance, there needs to be more discussion of both previous approaches and the advantages and disadvantages of the FAP-tagging approach as compared to those approaches. In particular, the need to genetically modify a cell line prior to use is a substantial disadvantage.

Also, the paper implies that endocytosis in their current cell line of choice (HeLa) is the same as all other cells, which the authors themselves have shown not to be true. For example, constitutive internalization of EGFR in mammary epithelial cells is quite distinct from most other cell lines, such as HeLa and fibroblasts (Burke and Wiley, 1999). EGFR endocytosis in mouse fibroblasts is also quite distinct from what is observed in HeLa and endothelial cells (Jiang et al., 2003). Thus, the approach outlined here would require tagging a variety of different cell lines to fully explore endocytic mechanisms and screen for regulators, which could be technically difficult.

It also has the disadvantage of being most useful for cells that express relatively low levels of receptors so that the signal from low levels of internalized receptors is not swamped out by high levels of receptors remaining on the cell surface.

Conversely, their approach has many advantages as compared to commonly used techniques. In particular, virtually all described methods for examining endocytosis of the EGFR are restricted to ligand-occupied receptors that behave dramatically different than empty receptors. In this respect, the technique is similar to the antibody-tagging technique (e.g. Burke, Schooler and Wiley, 2001), which has its own set of disadvantages, such as either blocking receptor occupancy (e.g. 225 mAb) or restricting the ligand which can be used (e.g. 13A9 mAb). However, radiolabeled antibodies have the advantage of yielding extremely high S/N ratios, thus also permitting evaluation of very low levels of receptor occupancy, even in cells with high levels of receptors.

---

## [Author Response]

Essential revisions:Please discuss the following issues, to place the work in better context, noting the comments made by the reviewers below.Please discuss the earlier work by others on the β2-adrenergic receptor more clearly, showing that the present work is an extension of work that had already developed the basic technology. We recognize that the advance here is to tag the endogenous receptor.

We expanded the discussion of the previous method development work by the Bruchez group. Please see insertions in the last paragraph of the Introduction and Results and Discussion, third paragraph.

Although this is clearly a very powerful technological advance, there needs to be more discussion of both previous approaches and the advantages and disadvantages of the FAP-tagging approach as compared to those approaches. In particular, the need to genetically modify a cell line prior to use is a substantial disadvantage.

We added the discussion of various other methods of studying membrane trafficking using pH-sensitive probes in comparison with the FAP-based technology. Please see Results and Discussion, fourth paragraph.

We agree with the reviewer that the necessity to label endogenous EGFR to implement our approach is a disadvantage. However, with the fast development of genome-engineering techniques, we think that tagging of endogenous proteins in multiple cell lines is becoming feasible. We included an additional comment discussing this point in the eleventh paragraph of the Results and Discussion.

Also, the paper implies that endocytosis in their current cell line of choice (HeLa) is the same as all other cells, which the authors themselves have shown not to be true. For example, constitutive internalization of EGFR in mammary epithelial cells is quite distinct from most other cell lines, such as HeLa and fibroblasts (Burke and Wiley, 1999). EGFR endocytosis in mouse fibroblasts is also quite distinct from what is observed in HeLa and endothelial cells (Jiang et al., 2003). Thus, the approach outlined here would require tagging a variety of different cell lines to fully explore endocytic mechanisms and screen for regulators, which could be technically difficult.

We certainly agree with the reviewer’s point. We included examples demonstrating the variability of endocytosis in various cell types. Here again, we think that studying endogenous proteins is highly important and feasible, especially in the case of EGFR, because the key tools to gene-edit EGFR and insert FAP are developed in our study and will be available. Please see Results and Discussion, eleventh paragraph.

It also has the disadvantage of being most useful for cells that express relatively low levels of receptors so that the signal from low levels of internalized receptors is not swamped out by high levels of receptors remaining on the cell surface.

We agree with the reviewer that the signal-to-noise ratio will be lower if measurements of the “whole-cell” 561/640 ratio are performed in cells expressing very high levels of FAP-EGFR. In fact, we have already generated FAP-EGFR-expressing gene-edited human squamous carcinoma HSC3 cells that express ~500,000 receptors per cell. Indeed, high-throughput screening for endocytosis regulators in the presence of 1 ng/ml EGF using the 561/640 nm ratio method in these cells was not as effective as in HeLa cells, although using 2 ng/ml resulted in statistically significant data. Therefore, we discuss a modified assay (see Results and Discussion, tenth paragraph) that we found to be more appropriate and reliable in cells like HSC3. New gene-edited HSC3 cell lines are currently in a process of final characterization in the laboratory, and therefore, we did not include experiments with these cells in the present manuscript.

Conversely, their approach has many advantages as compared to commonly used techniques. In particular, virtually all described methods for examining endocytosis of the EGFR are restricted to ligand-occupied receptors that behave dramatically different than empty receptors. In this respect, the technique is similar to the antibody-tagging technique (e.g. Burke, Schooler and Wiley, 2001), which has its own set of disadvantages, such as either blocking receptor occupancy (e.g. 225 mAb) or restricting the ligand which can be used (e.g. 13A9 mAb). However, radiolabeled antibodies have the advantage of yielding extremely high S/N ratios, thus also permitting evaluation of very low levels of receptor occupancy, even in cells with high levels of receptors.

We added a comparative discussion of an antibody-uptake endocytosis assay and FAP-based technology in the tenth paragraph of the Results and Discussion. Regarding the use of our method in cells with high levels of receptors, please, see our previous response.